# Study on the Structure and Dielectric Properties of Zeolite/LDPE Nanocomposite under Thermal Aging

**DOI:** 10.3390/polym12092108

**Published:** 2020-09-16

**Authors:** Bai Han, Chuqi Yin, Jiaxin Chang, Yu Pang, Penghao Lv, Wei Song, Xuan Wang

**Affiliations:** 1Key Laboratory of Engineering Dielectrics and Its Application, Ministry of Education, Harbin University of Science and Technology, Harbin 150080, China; ChuqiYinhust@163.com (C.Y.); 18845169304@163.com (J.C.); pangyuhust@163.com (Y.P.); lvpenghao95@163.com (P.L.); songwei791214@163.com (W.S.); topix@sina.com (X.W.); 2State Key Laboratory Breeding Base of Dielectrics Engineering, Harbin University of Science and Technology, Harbin 150080, China; 3College of Electrical & Electronic Engineer, Harbin University of Science and Technology, Harbin 150080, China

**Keywords:** polyethylene, zeolite, nanocomposite, thermal aging, breakdown, dielectric spectrum

## Abstract

Nanodoping is an effective way to improve the dielectric properties and the aging resistance of polyethylene. Nano-zeolite has a nano-level porous structure and larger specific surface area than ordinary nano-inorganic oxide, which can be used to improve dielectric properties of low-density polyethylene (LDPE) nanocomposite. The zeolite/LDPE nanocomposites were prepared and subjected to thermal aging treatment to obtain samples with different aging time. Using scanning electron microscopy (SEM), Fourier transform infrared spectroscopy (FTIR) and the differential scanning calorimetry (DSC) test to study the microscopic and structure characteristics, it was found that nano-zeolite doping can effectively reduce the thermal aging damage to the internal structure of the nanocomposite; carbonyl and hydroxyl decreased significantly during the thermal aging time, and the crystallinity effectively improved. Nano-zeolite doping significantly improved the morphology and strengthened the aging resistance of the nanocomposite. In the dielectric strength test, it was found that nanodoping can effectively improve the direct current (DC) and alternating current (AC) breakdown field strength and the stability after the thermal aging. The dielectric constant of nanocomposite can be reduced, and the dielectric loss had no obvious change during the aging process. Moreover, the zeolite/LDPE nanocomposite with the doping concentration of 1 wt % had the best performance, for the nano-zeolite was better dispersed.

## 1. Introduction

In the field of electrical insulation, especially for cable manufacturing industries, polyethylene is the most important power cable insulation material with the excellent insulating performance and mechanical properties [1,2]. In order to obtain better performances of polyethylene materials, many studies have tried doping polyethylene with nano-inorganic oxides. The dielectric properties of doped polyethylene are improved by using special effects under nanometer size [3,4,5] and interfacial effect between nanoparticles and polyethylene matrix particularly [6]. Various studies have shown that a small amount of nanodoping can significantly improve the dielectric properties of polyethylene, such as increasing its breakdown field strength, reducing its conductance under high electric field, reducing space charge accumulation and so on [7,8,9]. Common nanodopants are nano-inorganic oxides such as nano SiO_2_, Al_2_O_3_, MgO, TiO_2_, ZnO and so on [10,11,12,13,14]. Most of the studies have achieved some results and progress, but many problems remain to be solved. The microscopic mechanism of the modification of oxide nanoparticles is still at the level of qualitative description. It is generally believed that the interfacial region between nanoparticles and polymer matrix plays an important role, the special structure and dielectric behavior in the interfacial region are the key to the performance improvement [15,16]. Therefore, the mechanism of nanodoping modification is more likely to be the result of nanostructure in interfacial region rather than the nanoparticle itself, and the research on the polyethylene modification of nanodoping with special nanostructures from the perspective of microstructure has also been developed, such as the doping modification of nanoparticles with porous structure [17,18].

While studying the nanodoping modification of polyethylene materials, the aging of the insulating material research is also very important [19,20]. For high voltage and ultra-high voltage cables under the environment of high temperature and high electric field for long-term working, the insulation material is easy to accelerated aging and cause insulation failure [21,22]. Therefore, it is necessary to study the structure and dielectric properties of nanocomposite polyethylene after thermal aging. Some studies suggest that nanodoping can improve the thermal aging properties of polymer materials by improving the damage of molecular chains caused by thermal stress during thermal aging [23]. The reason is also related to the interaction between nanoparticles and polymer matrix [24]. In this paper, nano-zeolite polyethylene composites were prepared by using nano-zeolite with nano-porous structures as dopants and the aging characteristics of the composite materials were also studied. Nano-zeolites have a larger specific surface area than ordinary nanometer inorganic oxides and a special porous structure to enhance the interface effect in the nano-dielectric, so as to increase breakdown strength, reduce conductivity and suppress the space charge injection under high electric field, and also to enhance the thermal aging resistance of the nanocomposite. Under the condition of thermal aging, samples with different doping ratio and aging time were obtained, their structure and dielectric properties were studied, and the reason and mechanism of improving the aging performance were analyzed.

## 2. Materials and Methods

### 2.1. Materials

The basic material low-density polyethylene (LDPE) model NPC-LDPE was produced by Petrochemical Commercial Company of Iran (Tehran, Iran) and the density was 0.92 g/cm^3^. The type NaY zeolite was selected as nano scale dopant. The type NaY zeolite was produced by Tianjin Nanhua Catalyst Co. LTD and was prepared for laboratory use. The purity of nano zeolite was higher than 99.5%. The molecular formula of NaY zeolite nanoparticles is Na_56_[(AlO_2_)_56_(SiO_2_)_126_]•250H_2_O. Porous NaY zeolite with pore size about 0.74 nm and the particle size is about 50 nm. All materials were dried in an oven at 60 °C for more than 24 h prior to further blending.

Two kinds of sample with different mass fractions of NaY zeolite (1 wt % and 3 wt %) were prepared by melt blending with LDPE in a torque rheometer (Harbin Hapro Electrical Technology Co., LTD, Harbin, China) at 393 K for 10 min. Then the prepared nanocomposites were placed in flat vulcanizing machine (Huzhou Shuangli Automation Technology Equipment Co., LTD, Huzhou, China) at 393 K for about 15 min at 10 Mpa pressure to form films. The film thickness was 200 μm for break down test and 100 μm for dielectric spectrum and FTIR, respectively.

### 2.2. Aging Process

The oven with constant temperature was used for the thermal aging experiment, and 95 °C was selected as the temperature of thermal aging. During the thermal aging, the air blower was turned on to make the temperature in the oven uniform and keep the air composition consistent, so as to facilitate the aging oxidation reaction. In this paper, three kinds of materials were selected: pure LDPE, and type NaY zeolite/LDPE nanocomposites with nano-zeolite mass percentages of 1 wt % and 3 wt %. The pure LDPE and zeolite/LDPE nanocomposite performed four cycles of thermal aging (200, 400, 600, and 800 h, respectively) and each aging cycle was nearly an eight-day (200 h) time. All of the samples before and after aging were numbered, define Pure 0 to pure LDPE without aging, and Pure 1, Pure 2, Pure 3 and Pure 4 respectively to aging with 1 to 4 cycles; define NaY1-1 for zeolite/LDPE composites aging 1 cycle of nano zeolite doping concentration 1 wt %, similarly NaY1-2, NaY1-3 and NaY1-4 were zeolite/LDPE nanocomposites with aging 2 to 4 cycles respectively, and NaY3-1, NaY3-2, NaY3-3,Nay3-4 are zeolite/LDPE nanocomposites with nano zeolite doping concentration 3 wt % from 1 to 4 aging cycles, respectively.

### 2.3. Morphology and Structure Tests

The morphology of zeolite/LDPE nanocomposite was characterized by a Hitachi SU8020 scanning electron microscopy (SEM) (Hitachi High-Tech Co., LTD, Tokyo, Japan). Before the SEM test, the sample with a thickness of 200 μm was brittle fracture in liquid nitrogen, and the fracture surface of samples was gold-plated by E-1045 Ion Sputter instrument (Hitachi High-Tech Co. LTD, Tokyo, Japan) to improve the electrical conductivity of the sample and observed by SEM equipment subsequently.

Fourier transform infrared (FTIR) absorption spectra of pure LDPE and zeolite/LDPE nanocomposites films with 100 μm thickness were tested on a JASCO FT/IR-6100 spectrometer (JASCO Corporation, Tokyo, Japan). The tested wavenumber range was mid infrared spectroscopy (MIR) from 4000 to 400 cm^−1^ with a spectral resolution of 2 cm^−1^. The infrared absorption spectrum of the sample was obtained using transmission mode and each sample was scanned 5 times and averaged.

DSC-1 differential scanning calorimetry (DSC) analyzer manufactured by Mettler Toledo (Zurich, Switzerland) was used to test the absorption and release heat curves of the samples with variable temperature. All the samples were taken into small pieces with about 7 mg mass each, and the thermal analysis experiment was carried out in the atmosphere of nitrogen. The heating program was rising from 20 to 150 °C at 10 °C/min. The absorption and release heat curves were recorded and the heat absorbed during melting was calculated by integration. In order to retain the influence of the thermal aging process on the crystallinity of the sample, the heating curve and the value obtained during the first heating cycle were selected for the crystallinity calculation. Subsequently, the 100% melt enthalpy of PE was used to calculate the crystallinity of the sample. To ensure the accuracy of the experimental data, all the test samples were scanned twice and averaged.

### 2.4. Breakdown Test

The JNC801 type insulation dielectric strength tester (Shanghai Wang Xu Electirc Co., LTD, Shanghai, China) was used for DC and AC breakdown tests. Its output voltage range was AC 0~80 kV, and its boost rate was 2 kV/s. For DC breakdown test, half wave rectifier filter circuit is used for rectification. The test electrode was made of a ball - ball electrode with a diameter of 25 mm. In order to prevent surface discharge, the tested samples with average thickness of 200 μm and electrodes were immersed in transformer oil. DC and AC breakdown tests were carried out for 10 times for each sample and the data were recorded. Weibull distribution was used to calculate the breakdown characteristic function and breakdown field strength [25].

### 2.5. Dielectric Spectroscopy Test

Broadband Dielectric spectrometer models Alpha-A, the analyzer manufactured by Novocontrol Germany company (Frankfurt, Germany), was used for pure LDPE and zeolite/LDPE nanocomposite dielectric spectrum measurement. It can measure the frequency range from 3 × 10^−6^ Hz to 4 × 10^7^ Hz. The samples used for dielectric spectroscopy tests were 100 μm thick. Aluminum deposits were developed on the samples surface to obtain electrodes with a diameter of 25 mm by a vacuum coating machine (KYKY TECHNOLOGY CO., LTD., Beijing, China). And then, samples were placed in a constant temperature oven and dried at 60 °C for 24 h, and tested the dielectric spectrum with a frequency range of 10^−1^~10^7^ Hz. The relative dielectric constant and dielectric loss (tanδ) were calculated from the measured results.

## 3. Results and Discussion

Figure 1 showed the SEM images after the first and the fourth thermal aging cycle of different kinds of samples including pure polyethylene and zeolite/LDPE nanocomposite with nano-zeolite doped by 1 wt % and 3 wt %, respectively. SEM images can clearly show the microstructure changes of the samples during the thermal aging process. It can be seen from Figure 1a, for the pure LDPE after the first thermal aging cycle, the material internal began to produce tiny pore defects, and with the increase of aging period to fourth cycles shown in Figure 1b, pore defects increased and more obvious gaps on the cross section structure appeared. It manifested that thermal aging under aerobic conditions over time increasing made polyethylene material internal oxidation reacted, expand the size of the defects and quantity, and led to the decrease of the mechanical properties, cause on the cross section of broken gap deeper and more obvious.

However, the microstructure of the zeolite/LDPE nanocomposite was obviously different. It can be found that, from the image of zeolite/LDPE nanocomposite with zeolite doping concentration of 1 wt % shown in Figure 1c, the internal structure had no obvious change after the first thermal aging cycle and nanoparticles was relatively evenly dispersed in the nanocomposite. The size of all the nano-zeolite particles was below 100 nanometers, and some particles were aggregated together, but not closely. Hardly any pore defects were found, and there was no clear boundary line between nanoparticles and polyethylene matrix. When the aging time reached 4 cycles, no obvious pore defects were found inside the nanocomposite all the same from Figure 1d, but the fluctuation on the fracture section was more obvious.

Figure 1e was microscopic image of zeolite/LDPE nanocomposite with doping concentration of 3 wt % after the first thermal aging cycle, from the image the microscopic structure was similar with the sample of doping concentration 1 wt %, there was no apparent porosity defect of the sample, just more zeolite nanoparticles, and the size of nanoparticles was bigger. This suggested that when the doping concentration was high, part of the zeolite nanoparticles produced agglomeration phenomenon. When the aging time was 4 cycles, from Figure 1f it can be seen that pore defects similar to those in pure LDPE began to appear in the interior of the sample, but the quantity was less than pure LDPE. Moreover, the binding between large size nano-zeolite particles and the matrix was not tight enough, and there were obvious boundaries. It can be judged from the SEM images that the nano-zeolite doping effectively inhibits the oxidation and porosity generated in polyethylene during the thermal aging process, and the doping concentration of 1 wt % had the best aging resistance effect. Due to the porous structure of the nano-zeolite, the polyethylene molecular chain part may enter the porous structure, which made the polyethylene structure more stable, inhibiting the damage and defects in the process of thermal aging, and improving the anti-aging performance. When the zeolite doping concentration was 3 wt %, the size of the nanoparticles increases due to agglomeration, and the defects between the particles and the matrix increase, resulting in the deterioration of the aging resistance of the nanocomposite material.

The above phenomena were also verified by FTIR spectroscopy. Figure 2 shows the infrared absorption spectra of pure polyethylene and the zeolite/LDPE nanocomposite at 1% doping concentration at different aging cycles, respectively. As can be seen from Figure 2a, the absorption curve of FTIR spectrum showed all the characteristic absorption peaks of medium strength and strong strength of LDPE [26]. The absorption peak at 721 cm^−1^ identified the rocking motion of the larger atomic group of –(CH_2_)_n_– (n ≥ 4), the absorption peak located at 1460 cm^−1^ can also be derived from the scissor bending vibration of –CH_2_– or antisymmetric stretching vibration of –CH_3_. In addition, the absorption peak at 1367 cm^−1^ was also identified the symmetrical stretching vibration of –CH_3_. The wide peak near 2900 cm^−1^ was composed of two absorption peaks of 2850 and 2926 cm^−1^ [27], and was also the absorption peaks identified antisymmetric stretching vibration and the symmetrical stretching vibration of –CH_2_– respectively. This wide peak appeared to have been flattened, possibly because the intensity of the two absorption peaks for –CH_2_– was the strongest here which was completely absorbed by the sample and recorded with too high a concentration. It was worth noting that in the infrared spectrograph, there was an obvious absorption peak at 1712 cm^−1^ which is enhanced with the increase of the aging period from thermal aging cycles 1 to 4, and this absorption peak can be corresponding to the carbonyl (C=O) double bond stretching vibration absorption [28,29]. In addition, in the spectra curve with the aging cycles of 1, 3 and 4, it can also be seen that a wide absorption peak also appears at 3381 cm^−1^, which also increases with the aging time. This absorption peak can be corresponding to the stretching vibration absorption peak of hydroxy (–OH), or the frequency doubling stretching vibration absorption peak of C=O. Carbonyl and hydroxyl increasing with the thermal aging time indicated that the oxidation was reacting constantly in polyethylene at aerobic environment condition with the increasing of time. This oxidation caused structural defects of polyethylene molecules and increase the polarity of polyethylene materials, which will lead to the destruction of microstructure and the increase of pores, seriously affecting the dielectric properties of insulating materials finally.

Figure 2b shows the infrared spectra of zeolite/LDPE nanocomposites with doping concentration 1 wt % under different thermal aging cycles. The absorption peak of –CH_2_– and –CH_3_– was basically the same as that of pure LDPE. However, a wide absorption peak appears at 1037 cm^−1^, which can be considered as the characteristic peak of TO_4_ (oxide tetrahedron), was the characteristic peak of silicon-oxygen tetrahedron or aluminum-oxygen tetrahedron in the nano-zeolite. At the same time, significantly different from pure LDPE, the absorption peak of carbonyl (C=O) at 1712 cm^−1^ is significantly reduced, which is not obvious at 1 and 3 aging cycles, and can be seen only after the fourth aging cycle presenting a significant reduction compared with pure LDPE. In addition, the characteristic peak of hydroxy (–OH) located at 3381 cm^−1^ was also significantly weakened. The absorption peak could hardly be seen at 1 and 3 aging cycles, and it could only be observed at 4 aging cycles, which was consistent with the phenomenon of carbonyl reduction. These phenomena indicate that the addition of nano-zeolite significantly decreased the oxidative reaction under thermal aging in aerobic environment in polyethylene, inhibits the damage of oxidation when aging progress. At the same time, The Si and Al provided by nano-zeolite can form bond structure of C–O–Si or C–O–Al instead of carbonyl C=O in the matrix, and the change of chemical bond can reduce defects or improve the structural stability in polyethylene.

In order to further prove that zeolite/LDPE nanocomposites had better thermal aging resistance properties, a thermal analysis test was used to assist the proof. All samples were tested by differential scanning calorimetry (DSC), and the absorption curve was analyzed. A comparison was made between the 100% melting enthalpy value and the heat absorption during melting, so as to obtain the crystallinity and melt peak value (T_max_) information of all samples. The specific values are shown in Table 1. It can be seen from the table that the crystallinity (X_C_) of pure LDPE before aging is 37.72%, while the crystallinity of the zeolite/LDPE nanocomposite has been improved to 39.23% and 39.08% after doping with nano-zeolite. This indicates that nano-zeolite doping can effectively improve the crystallinity of the nanocomposite, increase the proportion of ordered structure inside the material, reduce the defects and pores caused by disordered structure, and thus improve the performance and aging stability of LDPE. After thermal aging from 1 to 4 cycles, the crystallinity of pure LDPE decreased slightly, indicating that with the increasing of aging time, the thermo-oxygen aging of polyethylene materials intensified, thermal cracking and thermo-oxygen degradation led to the fracture of LDPE macromolecular chains, partial crystal structure was destroyed, and the crystallinity decreased. However, the crystallinity of zeolite/LDPE nanocomposite was first increased and then decreased with the increasing of aging time. The increase of crystallinity after the first and second thermal aging cycles may be due to the fact that at the initial aging with temperature of 95 °C, recrystallization occurred to a small extent in the nanocomposite and nano-zeolite particles may be used as heterogeneous nucleation centers, resulting in the increasing of crystallization ratio and more regular and perfect material structure. When the aging time was further increased, thermal aging will also lead to oxidation damage inside the nanocomposite, thus reducing the crystallinity. However, the decrease of crystallinity in the nanocomposite was much less than that of pure LDPE, indicating that the aging damage degree level was lower than that of pure LDPE. Because of its porous structure, nano-zeolite may made part of polyethylene molecular chain enter the pore and more tightly bonded, which can stabilize the structure of composite material more effectively and increase the proportion of ordered structure. In addition, it was worth noting that in the DSC testing data, the melt peak value (T_max_) of pure LDPE presents the downward trend after rising first. However, the T_max_ of zeolite/LDPE nanocomposite appeared very stable, almost remaining the same in different thermal aging cycles; this suggested that the internal structure of the zeolite/LDPE nanocomposite remained stable after thermal aging, without apparent thermal degradation or thermal transitions [26,30].

The DC and AC breakdown Weibull distributions of all samples in different aging cycles were shown in Figure 3 and Figure 4, respectively. As the electrical breakdown was a random weak point breakdown process and the breakdown data has a certain dispersion, Weibull distribution was used to carry out statistics and analysis of the breakdown data. In Figure 3 and Figure 4, β is the shape parameter of Weibull distribution which reflects the dispersion degree of breakdown strength, while E_0_ is the breakdown field strength. As can be seen from Figure 3a, the DC breakdown field strength of all the zeolite/LDPE nanocomposites was higher than that of pure LDPE. Moreover, from Figure 3b–d the nano-zeolite doping effectively improved the DC breakdown field strength of the nanocomposites during the thermal aging process. With the increase of the aging cycle, the breakdown field intensity of all samples showed a downward trend, but the breakdown field strength performance of zeolite/LDPE nanocomposites decreased less than that of pure LDPE. After four rounds of thermal aging treatment, the breakdown field strength of pure LDPE decreased by 27.35% while that of zeolite/LDPE nanocomposite with nano-zeolite doping concentration 1 wt % decreased by 21.01%, which indicates that zeolite/LDPE nanocomposite has a better heat-resistant aging performance. The DC breakdown field strength of zeolite/LDPE nanocomposite with NaY nano-zeolite doping concentration 1 wt % was higher than that of the nanocomposite with NaY nano-zeolite doping concentration 3 wt % in all aging cycles, indicating that the sample with 1 wt % doping had the best DC breakdown performance. When the nano-zeolite doping concentration was high (3 wt %), the agglomeration of nanoparticles in nanocomposites was increased, the related defect and pore size would be larger, and large-size defects and destruction are more likely to occur during thermal aging process, which may lead to breakdown performance degradation and was consistent with the phenomenon of SEM and FTIR described above. The nano-zeolite with a large number of porous structures can effectively enhance the interface effect between nanoparticles and polyethylene matrix, so as to generate more and deeper traps in the interface area. The trapped charge under high electric field will block further charge injected, thus improving the electrical strength of the material.

As shown in Figure 4, the result of AC breakdown was basically consistent with that of DC breakdown. The breakdown field strength of zeolite/LDPE nanocomposite was higher than that of pure LDPE under different aging cycles. The aging resistance of zeolite/LDPE nanocomposite was better and the breakdown field strength decreased less. In addition, the shape parameters of zeolite/LDPE AC breakdown Weibull distribution are larger than that of pure LDPE, indicating that the nanocomposite had better AC breakdown stability. A large number of porous structures and enhanced interfacial effects brought by nano-zeolite doping can effectively reduce the electrical conductivity by trapping electrons, reduce the electron mobility and heat generation under the high Voltage of AC, and thus enhance the AC breakdown strength. Comparing between the nanocomposites with different doping contents, the AC breakdown field strength of nanocomposite with doping concentration of 1 wt % was still higher than that of nanocomposite with doping concentration of 3 wt % in all aging cycles which was also similar to the DC breakdown results. A higher doping concentration will lead to more agglomeration of nanoparticles within the nanocomposite, thus increasing internal defects and pores. In the process of thermal aging, it is more likely to generate weaknesses that lead to material breakdown under high electric field [31].

Figure 5 showed the dielectric constant and dielectric loss curve of pure LDPE under different aging cycles. It can be found that with the increase of aging cycle, the dielectric constant of the material shows an upward trend, increasing from around 2.2 to 2.5 in Figure 5a. It was because with the increase of aging time, some molecular chains inside the polyethylene material were destroyed, the number of polar groups increased, and at the same time, some polar groups located in the long molecular chain are detached, increasing the polarity of the material, resulting in the rise of the dielectric constant [20]. It can be seen from the dielectric loss diagram shown in Figure 5b that the total dielectric loss also increases with the increase of aging period. There is no obvious loss peak in the whole frequency range of Pure-1, indicating that the pure LDPE had no obvious dielectric relaxation even when it was suffered one circle of thermal aging. When the aging cycle increases to 2, 3, and 4, there were small dielectric loss peaks visible in the curve of samples Pure-2, Pure-3 and Pure-4 located at about 10~100 Hz. This dielectric loss peak may be related to the polar groups and defects in pure LDPE produced by the destruction of thermal aging. This was mainly because the polarization time range of the dipole moment generated by the defect and polar groups was 10^−6^–10^−2^ s, which corresponds to the frequency position of the current dielectric loss peak.

Figure 6 shows the dielectric constant and dielectric loss curves of the zeolite/LDPE nanocomposite with doping concentration of 1 wt % at different aging cycles. It can be found that, different from pure LDPE, with the increase of aging cycle, the dielectric constant decreases from around 2.4 to 2.3. This was mainly due to the nanoparticles doped inhibited the destruction of the thermal aging and at the same time small recrystallization happens inside, orderly nonpolar component increased, which can be proved from FTIR and DSC test conclusion. Moreover, there were a lot of porous structures and large specific surface area on the surface of nano-zeolite, the polyethylene molecular chain may enter into the porous structure of zeolite partly at a higher temperature, or bind more closely to reduce the polarity of the nanocomposite. It can be seen from Figure 6b that under different aging cycles, the dielectric loss did not change significantly, and there was a relatively obvious dielectric relaxation peak around 10^3^~10^4^Hz, which was mainly caused by the interfacial polarization between nano-zeolite and polyethylene. According to the theory of Maxwell–Wagner–Sillars, interface polarization relaxation time was relatively long, mainly occurred in the low frequency region about 10^−3^~10^3^ Hz [32]. However, due to the small size and the uniform dispersion of zeolite nanoparticles, the combination between nano particles and LDPE matrix was relatively close in zeolite/LDPE nanocomposites. Therefore, in the process of polarization and depolarization, the distance of charge movement was not as long as that of ordinary interfacial polarization in a larger scale, the polarization time was shorter, and the corresponding frequency of interfacial polarization was higher. This frequency could be even higher than the dipole polarization frequency by defects and polar groups in pure LDPE caused by thermal aging.

## 4. Conclusions

In this paper, morphology, infrared spectrum, crystallinity, DC and AC breakdown and dielectric spectrum of pure LDPE and zeolite/LDPE nanocomposites after thermal aging were investigated. The aging characteristics were fully studied from the aspects of microstructure and structural characteristics to macro dielectric strength and dielectric spectrum respectively, and the correlation and analysis were established.

The SEM microstructure characterization after aging showed that nano-zeolite doping could effectively reduce the damage of thermal aging on the internal structure of the nanocomposite, effectively reduce and hinder the generation of internal holes, and improve the heat-resistant aging performance. In the subsequent infrared spectrum test, the experimental results also verified that the carbonyl and hydroxyl groups in zeolite/LDPE nanocomposites were significantly reduced, and its characteristic peak was far lower than that of the pure LDPE with the same aging cycle. In DSC test, it was found that nano-zeolite doping could effectively improve the crystallinity of nanocomposites, and the crystallinity change was not obvious in the whole aging process, only slightly increased first and then decreased, which was significantly different from the monotonic decline of pure LDPE. The results showed that the nano-zeolite improved the structure characteristics inside the material, reduced the large size defects, and the nano-zeolite was more closely bound to the polyethylene matrix.

Through the test of DC and AC electric strength, it was found that the nano-zeolite doping can effectively improve the electrical strength of the nanocomposite, and the DC and AC electric strength are both improved in all aging cycles. Compared with pure LDPE, with the increase of aging time, the breakdown field strength decreases less, the stability is better, and the aging resistance is improved.

The dielectric constant of the zeolite/LDPE nanocomposite decreased slightly with the increase of aging time, which was contrary to the small increase of pure LDPE. At the same time, there was a loss peak of the nanocomposite under the action of interfacial polarization, but the loss was small and did not change significantly during the whole aging period.

The sample with a doping concentration of 1 wt % of nano-zeolite had better performance. When the doping concentration was increased to 3 wt %, the internal defects and pores of the material were larger due to the agglomeration of nanoparticles, which was more likely to cause damage during thermal aging.

## Figures and Tables

**Figure 1 polymers-12-02108-f001:**
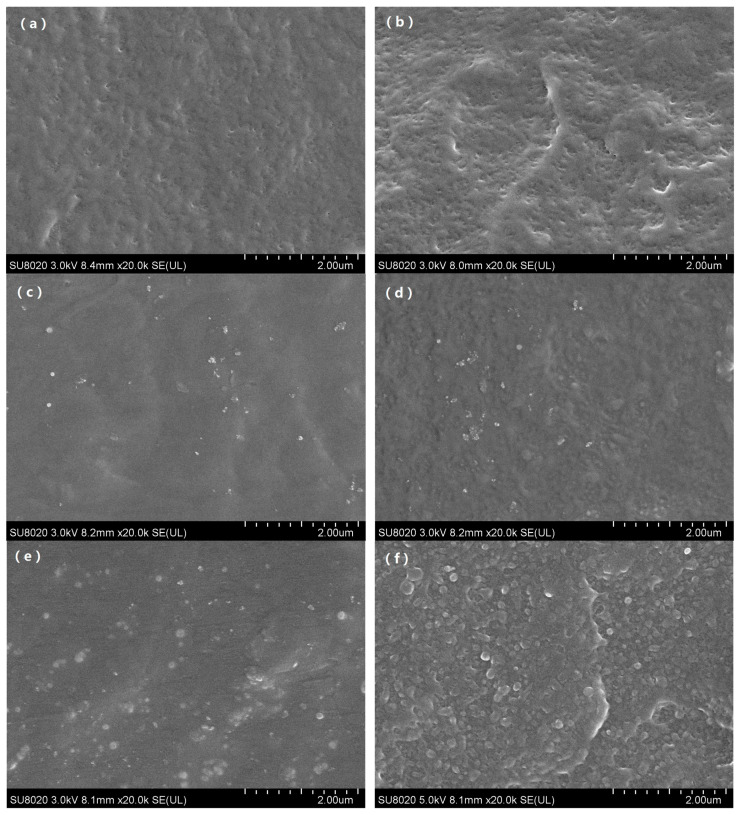
SEM image of pure low-density polyethylene (LDPE) pure 1 (**a**), pure 4 (**b**) and zeolite/LDPE nanocomposites NaY1-1 (**c**), NaY1-4 (**d**), NaY3-1 (**e**), NaY3-4 (**f**).

**Figure 2 polymers-12-02108-f002:**
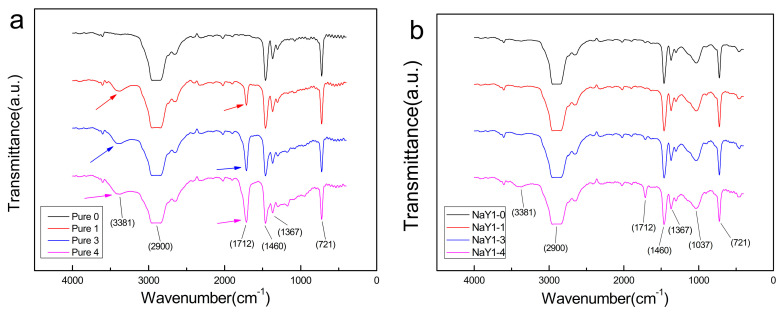
FTIR spectra of pure LDPE (**a**), zeolite/LDPE nanocomposite with nano-zeolite doping concentration 1 wt % (**b**) at 0, 1, 3 aging cycles.

**Figure 3 polymers-12-02108-f003:**
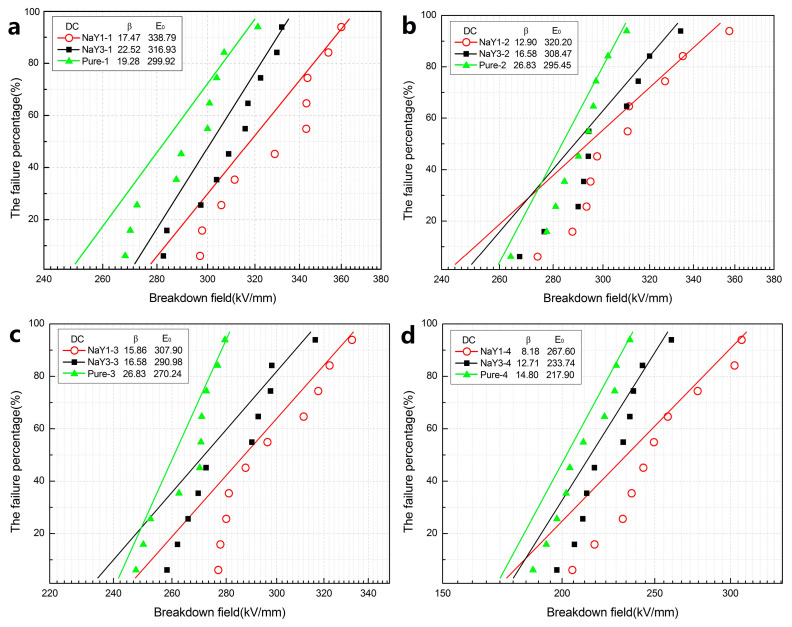
Weibull probability plots for direct current (DC) breakdown strengths of pure LDPE and zeolite/LDPE nanocomposites at 1 (**a**), 2 (**b**), 3 (**c**) and 4 (**d**) aging cycles.

**Figure 4 polymers-12-02108-f004:**
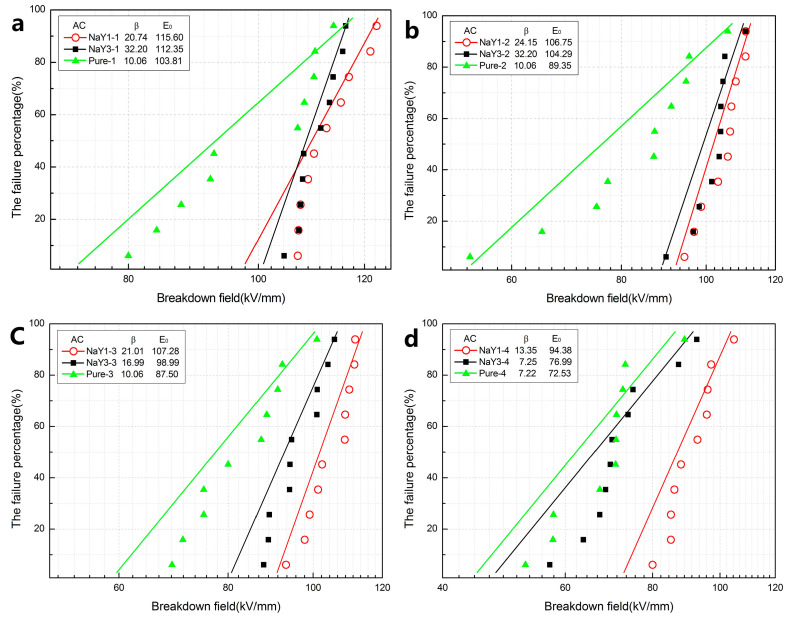
Weibull probability plots for alternating current (AC) breakdown strengths of pure LDPE and zeolite/LDPE nanocomposites at 1 (**a**), 2 (**b**), 3 (**c**) and 4 (**d**) aging cycles.

**Figure 5 polymers-12-02108-f005:**
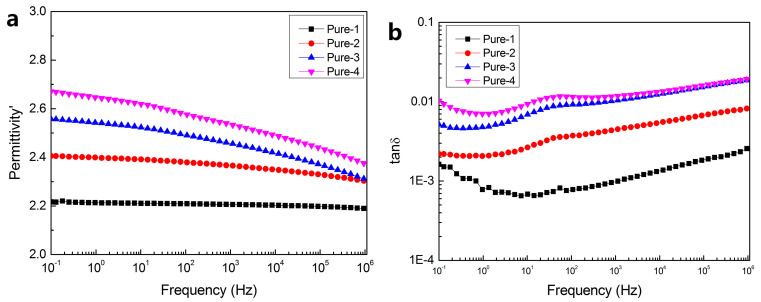
Dielectric spectrum of pure LDPE for dielectric constant (**a**) and dielectric loss (**b**).

**Figure 6 polymers-12-02108-f006:**
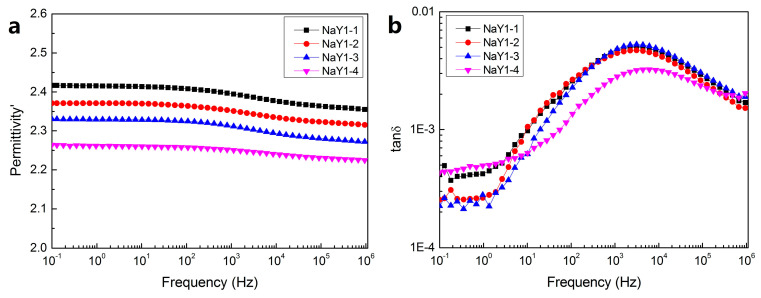
Dielectric spectrum of zeolite/LDPE nanocomposite with nano-zeolite doping concentration 1 wt % for dielectric constant (**a**) and dielectric loss (**b**).

**Table 1 polymers-12-02108-t001:** Crystallinity and melt peak value of pure LDPE and zeolite/LDPE nanocomposites at different thermal aging cycles.

Aging Cycles	0 Cycle		1 Cycle		2 Cycles		3 Cycles		4 Cycles	
Sample	X_C_ [%]	T_max_ [°C]	X_C_ [%]	T_max_ [°C]	X_C_ [%]	T_max_ [°C]	X_C_ [%]	T_max_ [°C]	X_C_ [%]	T_max_ [°C]
**Pure LDPE**	37.72	108.14	37.48	109.61	37.07	110.17	36.61	109.33	36.09	109.52
**NaY1**	39.23	109.13	39.12	109.83	40.93	109.51	40.63	109.05	40.25	109.06
**NaY3**	39.08	108.00	39.36	108.44	40.35	108.31	38.88	108.11	38.69	108.67

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
