# Peer review of "Study on the Structure and Dielectric Properties of Zeolite/LDPE Nanocomposite under Thermal Aging"

_polymers, 2020, doi:10.3390/polym12092108_

Round 1

Reviewer 1 Report

The paper is aimed to investigate the influence of thermal aging on the structural and electrical properties of zeolite/LDPE nanocomposite samples!

In respect to electric properties characterisation:

  1. Please do not use the term “better dielectric properties” (line 66). Please specify how you wanted to improve the dielectric properties: increased the value of dielectric constant or diminish it? There are applications where dielectric constant should be higher, there are other where dielectric constant should be lower!
  2. Lines 123 and 126: please modify the frequency values…I suppose the authors are referring to the freq domain .0.000001 Hz – 30MHz
  3. Instead of “Samples with a thickness of 100μm were evaporated with aluminum electrodes…”, I would reformulate and say that Al deposits were developed on the substrate’s surface. (Line 124)
  4. Line 247 “insulation breakdown” - this is not a consecrated term. Please use electrical or dielectric breakdown.
  5. Regarding Figure 5: there is a small peak visible in the case of samples Pure-2, Pure-3 and Pure-4. Which is its origin?
  6. I would revise the conclusion regarding the electrical properties. It is not clear if the authors measured the thickness of the samples before and after thermal aging? Which was the thickness before aging?
  7. Please verify in the literature case at 1kHz-10kHz the interfacial polarization is occurring. Normally, if it would appear at the frequencies you mentioned, it should not decrease while diminishing the frequency to 0.01Hz.

I do consider that the paper can be published only after performing major revisions.

Author Response

Dear Reviewer:

Your comments have been taken seriously and the following corrections to your comments were made to manuscript:

  1. Line 66-67: According to Reviewer’s suggestion, the description of the “better dielectric properties” has been changed to “increase breakdown strength, reduce conductivity and suppress the space charge injection under high electric field” to more accurately describe the specific content of improving the dielectric properties. For polymer insulating materials, there is no directly relationship between the dielectric constant and insulating property, so it is not intended to indicate whether the dielectric constant is increased or decreased.
  2. Line 132 and 136: There were typos in the digital superscript of the frequency range in the original manuscript, which have now been corrected. Other superscript problems in the original manuscript have also been corrected
  3. Line 130-137: According to your suggestion, the description of dielectric spectrum measurement is rewritten and the description of the closed aluminum electrode is more clearly.
  4. Line 272: “insulation breakdown” was replaced to “electrical breakdown”
  5. Line 333-340: In the original manuscript, the author did not pay enough attention to the small loss peak of pure LDPE after thermal aging in Figure 5. The author believes that this small loss peak may be related to the defects and polar groups caused by the internal damage of the material caused by thermal aging, which is explained in the revised manuscript.
  6. The reviewer mentioned some questions related to sample thickness in the review comments, for which we have the following explanation: In the original manuscript, the author did not describe the thickness of the thermal aging sample. After re-measurement and confirmation of the samples, we found that thermal aging did not change the thickness of the sample. In the process of sample preparation, there were mainly two kinds of thin film samples with thickness of 100 microns and 200 microns, respectively, with thickness deviation was about ±5%. After thermal aging, the thickness and shape of the sample did not change significantly, and the thickness was still within the deviation range. Since the thickness test is not a precise and convincing experiment, we did not include this part in the revised manuscript. If the reviewer consider it is necessary to explain, we will make further corrections.
  7. Line 351-361: The explanation of the frequency range of interfacial polarization is supplemented and perfected. This is mainly due to the small size and the uniform dispersion of zeolite nanoparticles, combination between nano particles and LDPE matrix was relatively close in zeolite/LDPE nanocomposites. Therefore, in the process of polarization and depolarization, the distance of charge movement was not as long as that of ordinary interfacial polarization in a larger scale, the polarization time was shorter, and the corresponding frequency of interfacial polarization was higher.

By taking your valuable advices, readers will be able to understand the research more clearly.

Authors of Polymers-925431

Reviewer 2 Report

Han and coworkers described study on the structure and dielectric properties of zeolite/LDPE nanocomposite. In the presented manuscript, morphology, infrared spectrum, crystallinity, DC and AC breakdown and dielectric spectrum of pure LDPE and zeolite/LDPE nanocomposite after thermal aging were investigated. The aging characteristics were studied. The SEM microstructure characterization after aging showed that nano-zeolite doping could effectively reduce the damage of thermal aging on the internal structure of the nanocomposite, effectively reduce and hinder the generation of internal holes, and improve the heat-resistant aging performance.

The aim of the article is important for material science, especially for research related to the thermal properties of materials however, some deficiencies exist, which are listed below.

-There are several typos, such as: “the density is 0.92 g/cm3”, “200Hours”, resolution of “2 cm-1” and many others

-Many sentence structures are incorrect, editing of English language and style is required.

-The materials and methods section is not sufficient prepared:

-Information on the origin and purity of the reagents used is missing, e.g. the origin of NaY, whether it was commercially available or obtained in a laboratory, what was the purity of it.

-No information was given whether the DSC value of the first or the second heating cycle was used. First heating is done to eliminate any thermal history that the polymer may have gone through during its synthesis and post processing steps.

-There is no information about what technique was used for the IR spectrum measurement, whether it was the pellet technique, nujol film, DRIFT or ATR.

-Authors also provided the information that: “Before the SEM test, the sample with a thickness of 200 m”, 200m cannot be a real value because it is more than the dimensions of the SEM apparat.

-Figure 2. The scale for the IR spectra is inappropriate, it should be in ascending order, starts from 4000 to 400, not 400 to 4000 cm-1.

FTIR analysis requires improvement, the peaks on the transmission spectrum in the infrared region do not correspond to a certain group but their vibrations, stretching, bending etc. For example for CH2: symmetric, antisymmetric, stretching, scissoring, rocking, wagging, and twisting. The fingerprint region (from about 1500 to 500 cm-1) usually contains a very complicated series of absorptions. These are mainly due to all manner of bending vibrations within the molecule rather than single group. Should not be considered as a single CH2 group vibration. The peaks in the 3000 to 2600 range look like they are clipped (they are flat) which is not characteristic of the stretching vibrations (symmetrical and asymmetric, characteristic for this range) of the CH2 group, the spectrum looks as it was recorded with too high a concentration or come from the used nujol (but no method of making these spectra is given, so this is only a speculation). In the discussion I am missing reference and discussion of major work on thermal properties of polymer materials, no reference to this substantial contribution to this area is given but furthermore, that research should be discussed in context to yours as it is highly relevant, for example: Polymer Degradation and Stability 2007, 92, 1721; New J. Chem., 2018, 42, 39; Thermochimica Acta 2006, 440, 36; J. Organomet. Chem. 2017, 847, 173, but many others are available. These publications can also help authors interpret their IR spectra.

Author Response

Dear Reviewer:

Your comments have been taken seriously and the following corrections to your comments were made to manuscript:  

There were some typos in the superscript marking in the original manuscript such as” g/cm3, cm-1, 10-6Hz, etc.”, which have now been corrected.

According to the reviewer’s suggestion, the English language writing of the manuscript was reviewed and reedited, and the structure of some sentences was revised.

The materials and methods section were modified and supplemented.

Line 73-78: The origin and purity of the materials (LDPE and nano zeolite) used are supplemented and explained.

Line 114-117: The data used in DSC and crystallinity calculation are described. In order to retain the influence of the thermal aging process on the crystallinity of the sample, the heating curve and the value obtained during the first heating cycle were selected for the crystallinity calculation.

Line 105-109: The methods and modes used in infrared testing are supplemented to make the description more clearly.

Line 101: The typo of “200 m” was corrected to “200 μm”

In Figure 2, the scale for the IR spectra was modified in ascending order, starts from 4000 cm-1 to 400 cm-1.

Line 186-196: The FTIR analysis was improved. The vibration modes corresponding to each infrared absorption peak were explained. The reason of the flattened peaks in the 3000 cm-1 to 2600 cm-1 range was explained. This wide peak appeared to have been flattened, possibly because intensity of the two absorption peaks for -CH2- was highest here and completely absorbed by the sample and recorded with too high a concentration. Moreover, references on IR spectra were supplemented. Ref[26] [27] [29]

Line 258-263: The discussion on thermal properties was supplemented. The additional analysis was mainly related to melting peak value analysis and related references Ref[26][30]. In this paper, only DSC was used to test the thermal property, and the DSC curve was not given. This was mainly because there was no other noteworthy phenomenon on the DSC curve except the melting peak, so only the crystallinity was calculated and the melting peak value was counted. One of the main contents of this paper, the structural characteristics after thermal aging, has been fully demonstrated and described by SEM and FTIR experiments. The main purpose of DSC experiment is to further prove the structural characteristics Therefore, we did not carry out more tests and analyses on thermal properties. We will continue to delve into thermal properties in future papers and we lack of experience in this field at present. We are very grateful for the references provided by the reviewer, and some of the references have been cited in the revised manuscript.

By taking your valuable advices, readers will be able to understand the research more clearly.

Authors of Polymers-925431

Round 2

Reviewer 2 Report

The authors have well addressed my comments and worked hard on their manuscript. It is now recommended the manuscript to be accepted as a publication in the journal. Congratulations to the authors.